# Spatio-Temporal Distribution of Aedes Albopictus and Culex Pipiens along an Urban-Natural Gradient in the Ventotene Island, Italy

**DOI:** 10.3390/ijerph17228300

**Published:** 2020-11-10

**Authors:** Mattia Manica, Sara Riello, Carolina Scagnolari, Beniamino Caputo

**Affiliations:** 1Department of Biodiversity and Molecular Ecology, Research and Innovation Centre, Fondazione Edmund Mach, 38010 San Michele all’Adige, Italy; 2Department of Public Health and Infectious Diseases, Sapienza University of Rome, Laboratory Affiliated to Istituto Pasteur Italia-Fondazione Cenci Bolognetti, 00185 Rome, Italy; sarariello@gmail.com (S.R.); beniamino.caputo@uniroma1.it (B.C.); 3Department of Molecular Medicine, Laboratory of Virology, Sapienza University of Rome, Laboratory Affiliated to Istituto Pasteur Italia-Fondazione Cenci Bolognetti, 00185 Rome, Italy; carolina.scagnolari@uniroma1.it

**Keywords:** *Aedes albopictus*, *Culex pipiens*, spatial distribution, public health, zoonoses, Ventotene island, pathogen transmission, urban ecology, natural to urban gradient

## Abstract

The distribution of mosquitos and their corresponding hosts is critical in public health to determine the risk of transmission for vector-borne diseases. In this pilot study conducted in the small Mediterranean island of Ventotene, a known stopover site for migratory birds, the spatio-temporal distribution of two major mosquito vectors is analyzed from the natural to urban environment. The results show that *Aedes albopictus* aggregates mostly near areas with a human presence and the urban landscape, while *Culex pipiens* is more spatio-temporally spread, as it can also be found in wilder and less anthropized areas where the availability of human hosts is limited. *Culex pipiens* is also active earlier in the year. From a microgeographical perspective, our results confirm the anthropophilic spatial distribution of *Ae. Albopictus*, while suggesting that the circulation of bird zoonosis, such as West Nile, could be favored by the *Cx. pipiens* distribution. The results highlight the different ecology of the vectors and the interplay with their hosts, even at a small scale. The current evidence may help in forecasting the risk of pathogen transmission and surveillance planning.

## 1. Introduction

The presence of invasive mosquito species, such as *Aedes albopictus* [1] as well as native species like *Culex pipiens* [2], is of public health relevance [3,4] and cannot be underestimated. Both species are competent vectors of many pathogens that can greatly affect the health of both individuals (by developing severe symptoms) and communities (by blocking blood donations). However, even if the transmission route of the pathogen via the mosquito vector is known, the ability of the public health authorities and scientific community to forecast and understand the risks of transmission remains still limited [5]. Indeed, it remains yet to be fully understood whether the mechanisms and conditions that caused the extraordinary increase in West Nile virus cases in Europe during 2018 [6,7] and triggered the chikungunya outbreak in 2017 [8] were exceptional or potentially forecastable. Among other variables, the spatial distribution and seasonality of vectors, as well as their interconnections with hosts, are critical factors that can greatly impact the risk of transmission [9].Therefore, it is essential to understand how these species are distributed across the landscape and what natural and anthropogenic factors influence their spatial distribution.

Islands can be key observational sites for investigating the public health implications of such a tight coexistence between host and vector and for defining urban and landscape planning strategies to reduce human exposure to pathogens while preserving habitats. Studies have shown that Mediterranean islands feature a series of continuous and often competing interactions among natural and anthropogenic processes [10]. In recent years, tourism development has affected most small Mediterranean islands [11], thereby increasing the flow of people and goods. Moreover, many of these Mediterranean islands are initial stopover sites for migratory birds flying considerable distances [12]. The island of Ventotene is part of the Pontine Islands in the Tyrrhenian Sea off the Italian coast. It is a small island (124 hectars) stretching for less than 3 km along a NE–SW direction with a maximum width of less than 800 m. Ventotene Island is on the migration routes of many bird species that pass through the whole territory and are, therefore, recorded on the island during their different migration periods. Human density on the island is distributed along a NE to SW transect, from the touristic port and city to the southern natural reserve. A community of about 800 permanent residents lives on the island, but the population dramatically increases due to tourism during the summer months. The island is administratively part of the region Lazio, where 402 confirmed and probable human cases of chikungunya [13] and three equine cases of West Nile Virus [14] were recorded in 2017.

In this study, we placed 10 adult traps for sampling host seeking vectors along a SW to NE transect, from the southern natural reserve to the highly urbanized area inside the town, to evaluate the spatial distribution of the two main mosquito vectors in Italy, *Ae. albopictus* and *Cx. pipiens*. This allowed us to better understand mosquito and human interactions across the landscape and the potential exposure to zoonotic diseases.

## 2. Materials and Methods

Adult mosquito abundance was evaluated using 10 BG-Sentinel traps (BGs, Biogents AG, Regensburg, Germany) homogeneously distributed along a transect encompassing the entire length of the Ventotene Island (GPS: 40.793117, 13.426610), as shown in Figure 1a. The transect of the trap from the south-western natural reserve to the highly urbanized northeastern area inside the town follows the urbanization process on the island. The distance between the two BGs at opposite ends was 2.17 km, while the minimum distance was 161 m, and the average distance between the nearest traps was 265 m. Each BG trap was equipped with carbon dioxide and powered by a battery (12V 12AH art. consip batt-lead-13, Matsuyama Co., Ltd., Tokyo, Japan) activated for 24 h during 14 collections (from 10 March to 7 September 2014). To quantify the urbanization gradient, we computed the percentage of the area covered by buildings around each trap in a 250 m buffer (Appendix A). We chose the radius of the buffer based on the average distance between the nearest traps and the estimated flight range of *Ae. albopictus* [15]. We also explored how the choice of radius may have affected the analysis by testing a regular sequence of 10 radii from 50 to 500 m.

### Statistical Analysis

The spatial distribution along the urbanization gradient of adult *Ae. albopictus* and *Cx. pipiens* abundance was analyzed by means of a Generalized Additive Model (GAM) with Negative Binomial distribution and a log link. Poisson’s distribution was initially considered but excluded due to overdispersion. The dependent variable was the counts of mosquitoes trapped in each trap during the weekly collection. The independent variables included in the model were the mosquito species, the percentage of area covered by buildings, and their interactions. The week of collection was also included as a smoother function to model the temporal dynamics of both species separately. We used a thin plate regression spline to model the smoother term. We assessed whether the statistical assumptions of the model were met via a graphical inspection of the residuals. Given the GAM estimated parameters, it was possible to calculate the mean abundance of trapped mosquitoes and the probability of observing at least one capture (detection probability) to produce an estimate of the beginning of the seasonal activity of the two mosquito species. All statistical analyses were carried out using the R software version 4.0.2 (R Core Team, Vienna, Austria) [16] and the mgcv packages [17]. The R code and data are available online [18]. 

## 3. Results

Overall, we collected 121 *Ae. albopictus* and 539 *Cx. pipiens* specimens with an average of 0.864 (SD = 2.27) and of 3.850 (SD = 7.78) mosquitos/traps/collections, respectively (Figure 1b,c). In addition, during sampling, we collected three other mosquito species: Two species belonged to the genus *Culiseta* (i.e., *Culiseta annulata, Culiseta longiareolata*) and one to the genus *Aedes* (i.e., *Aedes mariae*).

The results of the model showed that *Ae. albopictus* is unevenly distributed within the island (Table 1, Figure 2), with an average abundance 8.1 higher in the urban area compared to the natural area. On the other hand, *Cx. pipiens* was more widespread on the island, also presenting a higher average abundance but lacking any meaningful statistical association with the percentage of area covered by buildings (Table 1, Figure 2, Appendix A). Changing the radius of the buffer affected the results only when radii lower than 200 m were considered. Specifically, the parameters for the percentages of buildings and their interactions with species were not statistically significant at a 0.05 threshold, even if their signs remained the same (see Appendix A and the available code for further details).

The seasonal patterns indicated that the seasonal activity of *Cx. pipiens* started earlier in the season (Figure 2, Appendix A) and then continuously increased, with the exception of week 32 (early August), when a control intervention based on adulticide sprayings (planned and carried out by local stakeholders independently from the present study) was carried out in the urban area. Interestingly, this intervention seemed to have only temporarily affected the mosquito abundance and dynamics, with the *Cx. pipiens* abundance quickly recovering to pre-intervention levels in about 3 weeks (Figure 1b). On the other hand, the population dynamics of *Ae. albopictus* post-intervention showed neither an increase nor a sharp decrease in the following weeks post-treatment (Figure 1b).

The estimated detection probability, obtained from the probability of observing at least one capture using fitted Negative Binomial distribution, showed that *Cx. pipiens* was active earlier in the season compared to *Ae. albopictus,* and only by week 27 (end of June) and in the most urban area was the detection probability of the latter comparable to the former (Figure 3). Indeed, *Ae. albopictus* was first detected by a single trap capture in site 7 during week 21 (end of May), while *Cx. pipiens* was detected earlier during week 18 (at the beginning of May) at the same site.

## 4. Discussion

In this pilot study, we presented results that consolidate the current knowledge regarding *Ae. albopictus* and *Cx. pipiens* ecology. In our field work, by deploying adult traps along a natural to urban gradient, we demonstrated that even with the limited size and resources of a small island, these two mosquito species show peculiar spatial and temporal patterns. The results confirmed the assumption of a greater abundance of *Ae. albopictus* in the urban area of the island. Indeed, the human-host preference of *Ae. albopictus* and its ability to exploit available man-made breeding sites are well known [1]. This species’ opportunistic behavior and great adaptability are at the roots of its success as an invasive species [19]. However, being able to observe differences in this species’ distribution even at this small a scale (less than 3 km length) highlight the critical role of the landscape ecotone. The observed field captures suggest that *Ae. albopictus* likely colonized the entire island, as it was detected at both ends of the gradients, yet adults tend to aggregate in specific hot-spots. Targeting these hot-spots may prove to be an effective strategy to reduce *Ae. albopictus* infestations on the whole island [20]. It should be noted that *Ae. albopictus* benefits from passive dispersal due to human mobility (e.g., cars, planes, and ships) [21]. However, the circulation of cars is restricted on the island, and there is only one possible point of entry (the port). Therefore, the characteristics of the land cover and land use could have shaped the spatial distribution of *Ae. albopictus* within the island while also considering the contrasting evidence on the actual *Ae. albopictus* flight range (about 250 m [15] or about 850 m [22]).

For *Cx. pipiens*, the results show a greater and more homogenous spread across the island, with aggregations in both natural and urban areas. *Cx. pipiens* is known to be a species complex (*Culex pipiens pipiens* and *Culex pipiens molestus*) with host preferences ranging from birds to humans [23]. Unlike *Ae. albopictus*, *Cx. pipiens* is an autochthonous species with a greater flight range [24], which may also explain its wider presence on the island. The population dynamics of these two species showed a delay at the start of the season for *Ae. albopictus* compared to *Cx. pipiens*. Such dynamics have been already observed in mainland Italy [25] and were confirmed in the current study. However, in our study, *Cx. Pipiens* was more abundant, suggesting that this species is better adapted to the ecological niche of the island and that interspecific competition may be reduced by a low abundance of *Ae. albopictus* and a different spatial distribution of the two species. Moreover, the distribution of *Cx. pipiens* over the whole island may also have played a significant role in the species’ ability to recover from the adulticide spraying carried out in the urban area. These types of control interventions are often carried out in touristic area in Italy to reduce the mosquito nuisance, even if the national guidelines recommend to limit their use only to exceptional situation or to disrupt the transmission of pathogens.

Pathogen transmission results from the presence of a competent vector, such as both *Ae. albopictus* and *Cx. pipiens*, and the presence or introduction of an infected host. Ventotene, a stop-over site along many migratory routes and a well-known touristic destination, could be an ideal site for targeted surveillance [26] and highlights the importance of further investigating the interplay and synchronization between the seasonal dynamics of both vectors and human tourism and bird migration. Unfortunately, the surveillance of all the pathogens currently circulating in the wild is challenging [27], and many arboviruses could emerge in the future and become a threat to public health [28]. West Nile, Usutu, and chikungunya are some of the viruses that are circulating or have been detected in the nearby central part of Italy [14,29] and could be easily introduced to the island by migrant birds or tourists. At present, the most common strategy to reduce transmission risk is by reducing vector abundance. However, there is a growing need for stronger evidence supporting the impact of vector-control in reducing West Nile transmission [30] in the face of the increasing costs of vector control. The lack of action thresholds and public health outcome assessments are also major drawbacks in optimizing vector control [31]. Preventive interventions based on the conservation of green spaces, human awareness, and the usage of biological larviciding products are increasingly recommended over control strategies that have a greater environmental impact (e.g., insecticide spraying), particularly for the control of *Ae. albopictus* [32]. Indeed, vector control strategies need to address the challenge of not altering natural processes, leaving a small environmental footprint, and avoiding the development of resistance to insecticides [33]. In the present times, when the harmonization of surveillance and control interventions and the sharing of expertise are advocated all over Europe [32,34], these results suggest that tailored surveillance should account for small-scale spatial heterogeneity and include areas exposed to pathogens introduced by non-human sources. In particular, the implementation of a wide-scale low density trap network could be useful for the surveillance of invasive mosquito species. Such a strategy would alleviate the limitations imposed by available finances, although it may fail to capture hot-spots of mosquito abundance and the risk of pathogen transmission.

## 5. Conclusions

The present study showed that mosquito populations are characterized by a heterogeneous spatial distribution, even at scale that is beyond the current sustainable entomological surveillance. These findings further stress the need to understand the spatial structuring of mosquito species related to ecological and human processes to better assess the risk of pathogen transmission. Indeed, both *Ae. albopictus* and *Cx. pipiens* are competent vectors and act as a bridge for the transmission of zoonosis between humans and animals. Therefore, we suggest that the planning of entomological surveillance and control interventions to be mandatory. Moreover, landscape and urban ecology could give insight into the underlying mechanisms characterizing mosquito distribution and their potential exposure to infected hosts, which, in turn, is essential for any control strategy and has great implications for public health.

## Figures and Tables

**Figure 1 ijerph-17-08300-f001:**
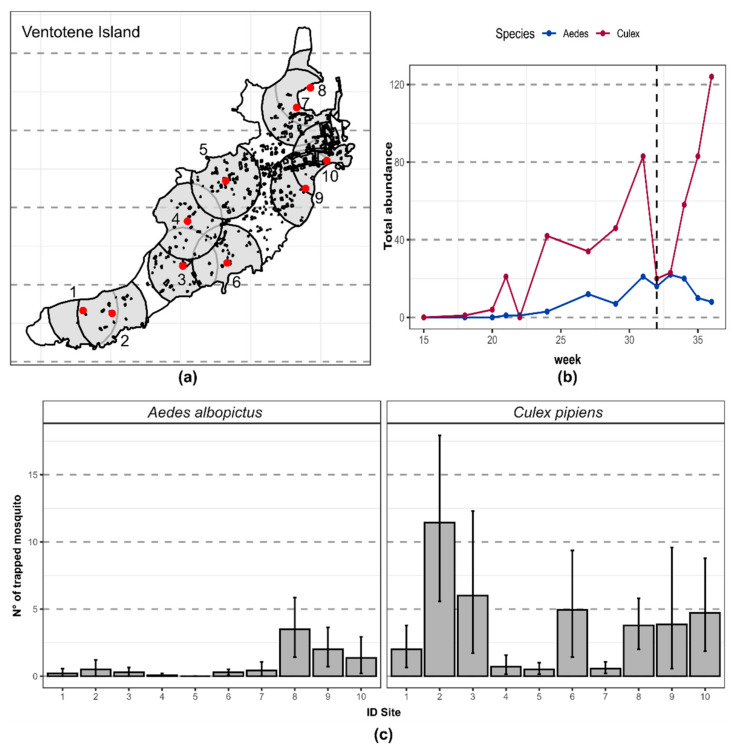
(**a**) Ventotene Island. The red dots are the BG-traps; the number uniquely identifies them along the natural to urban gradient, quantified by the percentage of area covered by buildings (black dots) in a 250 m radius buffer around each trap (grey area). (**b**) The total number of mosquitoes collected per species (*Aedes albopictus, Culex pipiens*). On the x-axis, the week of collection is included, while the y-axis presents the total number of mosquitoes captured. The dots represent the week of sampling. The vertical dashed line represents a pest control intervention using insecticide spraying on the island. (**c**) Average number of mosquitoes captured per trap. The x-axis presents the site identification number; see panel (**a**). The y-axis shows the number of mosquitoes captured. Bars represent the average number, and vertical lines represent a 95% confidence interval of the mean.

**Figure 2 ijerph-17-08300-f002:**
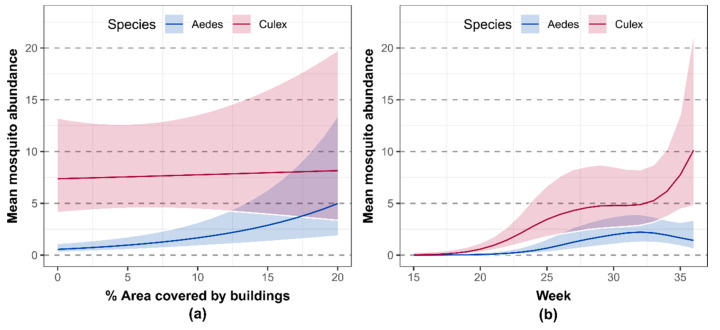
Results of the generalized additive model: (**a**) Relationship between the percentage of area covered by buildings (natural to urban gradient) and the mosquito abundance. The x-axis represents the percentage of area covered by buildings, and the y-axis represents the mosquito abundance. The solid line represents the average mosquito abundance, and the shaded areas are the 95% confidence intervals; (**b**) seasonal dynamics of mosquito abundance as estimated by the smoothers in the generalized additive model. The x-axis represents the week of the year, and the y-axis represents the mosquito abundance. The solid line represents the average mosquito abundance, and the shaded areas are the 95% confidence intervals. The average was computed assuming a 10% area covered by buildings.

**Figure 3 ijerph-17-08300-f003:**
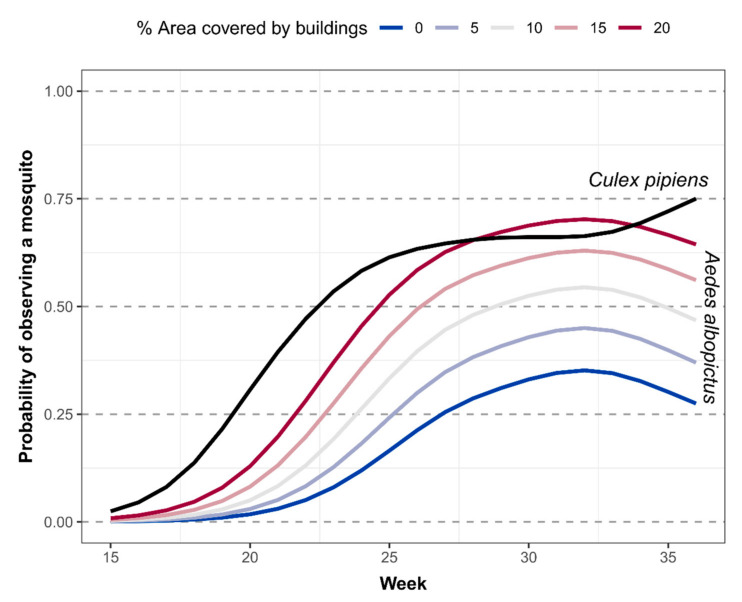
Detection probability of mosquitoes in the 0–1 range. The x-axis shows the week, and the y-axis shows the probability of observing at least one mosquito. The black solid line represents the detection probability of *Culex pipiens*, which is not statistically associated with the percentage of area covered by buildings. The colored solid lines represent the detection probability of *Aedes albopictus* conditional on the percentage of area covered by buildings.

**Table 1 ijerph-17-08300-t001:** Results of the generalized additive model. The dispersion parameter of the Negative Binomial distribution was estimated as 0.435 (*f()* indicates the smother (thin plate regression) spline).

Parameter	Estimate	Std. Error	Z Value	Pr.(>|z|)
Intercept	−2.015	0.429	−4.702	<0.0001
Species (*Culex pipiens*)	2.634	0.486	5.416	<0.0001
% buildings	0.105	0.029	3.680	0.0002
% buildings × Species	−0.099	0.038	−2.642	0.0082
*f*(week, *Aedes*)				0.0003
*f*(week, *Culex*)				<0.0001

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
