# Peer review of "Spatio-Temporal Distribution of Aedes Albopictus and Culex Pipiens along an Urban-Natural Gradient in the Ventotene Island, Italy"

_ijerph, 2020, doi:10.3390/ijerph17228300_

Round 1
Reviewer 1 Report
Manica et al. present an interesting study focused on the distribution of two mosquito species across an island: Ae. albopictus, an invasive species, and Cx. pipiens, an autochthonous species. The geographical and anthropic characteristics of the island were well used during the study to control for as many factors as possible and account for the ecological variations in both insects. Furthermore, models explaining the distribution of the mosquitoes in function of the urban density are presented and well described. This report presents a strong framework that can be replicated in multiple locations for unveiling ecological characteristics on zoonotic vectors and better plan for their control. Following some small suggestions that I think could help to improve the presentation of the results.
1-Perhaps, adding two map colored according to the distribution of the total number of mosquitoes per species could help to visualize the main conclusions of the study.
2-It is worth to clarify whether the pest control intervention was applied over the whole island or in particular regions of the island. In the latter case, this information can be shown in the maps suggested in the point above.
3-In lines 112, 114 and other points in the text, consider replacing "%" for the actual word "percentage", or better refer to the measure as urban density or building density.
4-In figures 1b, 2b and 3, consider adding below the week numbers the approximate corresponding months for easy interpretation of the recorded data.
5-Consider adding a chart displaying the average temperature and rain precipitation across the weeks to provide another dimension to the context of the study and interpretation to the meaning of Cx. pipiens becoming active earlier.
6-Line 153, edit "shows" to "show".
Reviewer 2 Report
I have reviewed the manuscript titled “Spatio-temporal distribution of Aedes albopictus and Culex pipiens along an urban-natural gradient in the Ventotene Island, Italy” by Manica et al. The study reports on a 14 sampling occasion survey of mosquitoes using a network of 10 BGs traps. Studies of this nature are useful in reporting the relative abundance and diversity of mosquitoes within a local area and may assist with local health authorities manage mosquito risk.
The study is relatively straight forward and while there may have been greater detail on spatial mosquito populations with a higher number of trap sites operated, I appreciate that there often operational constraints in this type of study.
The objective of assessing the local landscape and its influence on mosquito abundance is important. However, these two species, while having some preferences for the type of habitat used for oviposition and larval development, are generally both considered mosquitoes of urban landscapes. If this is not the case in the local circumstances, authors should outline this in introduction. Subsequently, the insights arising from the analysis require some additional commentary. I would liek to see more explanation of the local habitats and differences between sites 1-3 and 8-9 and the resulting relatively difference between the two species. More specific information is required on the habitats likely to be used by these species across the study site. For example, what is the "ecological niche" of the island as it specifically related to each species?
I have a number of questions for the authors.
- Were there only two species of mosquito collected? How does that compare to any other known lists of species from the local area. Did the trap styles used in this study bias collections to these species.
- What was the influence of prevailing climatic/environmental conditions on the local mosquito populations?
- The authors could include more information on the control intervention undertaken, was it targeting Aedes albopictus or Culex pipiens?
- What do the authors suggest the implications of this study would be for other exotic mosquitoes of concern in Europe. Would they make any recommendations on revise trap network or specific issues related to trap placement?
The manuscript does need a thorough revision to address some issues relating to grammar and general formatting (e.g. italics of scientific names).
